# Factors for the Early Revision of Misdiagnosed Tuberculosis to Lung Cancer: A Multicenter Study in A Tuberculosis-Prevalent Area

**DOI:** 10.3390/jcm8050700

**Published:** 2019-05-17

**Authors:** Chin-Chung Shu, Shih-Chieh Chang, Yi-Chun Lai, Cheng-Yu Chang, Yu-Feng Wei, Chung-Yu Chen

**Affiliations:** 1Department of Internal Medicine, National Taiwan University Hospital, Taipei 100, Taiwan; ccshu@ntu.edu.tw; 2College of Medicine, National Taiwan University, Taipei 100, Taiwan; 3Department of Internal Medicine, National Yang-Ming University Hospital, Yilan County 260, Taiwan; 11319@ymuh.ym.edu.tw (S.-C.C.); toto881049@yahoo.com.tw (Y.-C.L.); 4Division of Pulmonary Medicine, Department of Internal Medicine, Far Eastern Memorial Hospital, New Taipei City 220, Taiwan; koala2716@hotmail.com; 5Division of Chest Medicine, Department of Internal Medicine, E-Da Hospital/I-Shou University, Kaohsiung 824, Taiwan; yufeng528@gmail.com; 6Institute of Biotechnology and Chemical Engineering, I-Shou University, Kaohsiung 824, Taiwan; 7Division of Pulmonary and Critical Care Medicine, Department of Internal Medicine, National Taiwan University Hospital Yunlin Branch, Yunlin County 640, Taiwan

**Keywords:** tuberculosis, lung cancer, misdiagnosis, invasive procedure, revising

## Abstract

Background: Lung cancer misdiagnosed as tuberculosis (TB) is not rare, but the factors associated with early diagnosis revision remain unclear. Methods: We screened the cases with TB notification from 2007 to 2018 and reviewed those with misdiagnosis with a revised diagnosis to lung cancer. We analyzed the factors associated with early diagnosis revision (≤1 months) and early obtained pathology (≤1 months) using multivariable Cox regression. Results: During the study period, 45 (0.7%) of 6683 patients were initially notified as having TB, but later diagnosed with lung cancer. The reasons for the original impression of TB were mostly due to image suspicion (51%) and positive sputum acid-fast stain (AFS) (27%). Using multivariable Cox proportional regression, early diagnosis revision was associated with obtaining the pathology early, lack of anti-TB treatment, and negative sputum AFS. Furthermore, the predictors for early obtained pathology included large lesion size (>3 cm), presence of a miliary radiological pattern, no anti-TB treatment, and a culture-negative result when testing for nontuberculous mycobacteria (NTM) using multivariable Cox regression. Conclusion: In patients who are suspected to have TB but no mycobacterial evidence is present, lung cancer should be kept in mind and pathology needs to be obtained early, especially for those with small lesions, radiological findings other than the miliary pattern, and a culture positive for NTM.

## 1. Introduction

Tuberculosis (TB) remains the most common infectious disease worldwide [1] and, according to the World Health Organization (WHO), an estimated 10.0 million people had active TB, with 1.3 million TB-related deaths reported in 2017 globally [2,3]. Diagnosis optimization is key to providing patient care [4,5]. Although the diagnosis tools regarding TB have been improved in recent decades [6], there are still many cases diagnosed using clinical suspicion without positive culture evidence [7,8,9,10] due to time constraints [11]. The most common reasons for a diagnosis based on clinical suspicion are the radiographical or pathological presence of a lesion plus the difficulty and high risk of sample collection other than sputum, or the presence of an acute illness in which TB is suspected without improvement with broad-spectrum antibiotics [7]. In these cases, we might suggest empirical TB treatment and closely follow-up the treatment response. Once a positive treatment response is achieved, the diagnosis of TB can be suggested [7]. However, the accuracy of the tentative diagnosis of TB is still imperfect, with some diagnoses inevitably being incorrect.

Among patients with misdiagnosis, lung cancer is a concern because of its increasing incidence and high mortality [12,13]. In addition, both TB and lung cancer have similar pulmonary manifestations [14,15], such as cavitary lesion, miliary pattern, and pleural effusion [16,17], which can be a trap for clinicians to make a wrong diagnosis. Taiwan has an intermediate prevalence of TB. Differentiation of pulmonary TB from lung cancer can pose a great challenge to clinicians. However, there have been few studies on the timing of invasive diagnostic procedures for diagnosis of pulmonary TB, especially in TB-endemic areas. In a previous study, 1.87% of initial TB notifications are finally diagnosed as lung cancer instead [18]. The disease progression and mortality rate may be increased due to the delay in the cancer diagnosis [19,20,21]. In addition, TB treatment could induce adverse events in this population with initial misdiagnosis [7,22]. Therefore, in these situations, revising diagnoses early is important, but unclear. We conducted this study to review the patients with an initial notification of TB and then a revised diagnosis to lung cancer in a TB-prevalent area. We aimed to analyze the factors associated with early revision that could improve current patient care. 

## 2. Methods

### 2.1. Participant Enrollment 

This retrospective study was conducted in multiple tertiary referral centers in Taiwan, with one center located on each of the northern, middle, southern, and eastern sides of Taiwan, respectively. Under the approval of the Institutional Review Board of Research Ethics Committee of the study hospitals (NO. 201811047RINA), we screened patients aged ≥20 years that had been diagnosed with tuberculosis by formal notification to Taiwan Centers for Disease Control from January 2007 to August 2018 (details are in the Appendix A). We identified the patients with a final diagnosis of non-TB and among them, newly diagnosised lung cancer responsible for the pulmonary lesion. The patients with pre-existing lung cancer, co-existence of TB and lung cancer, and human immunodeficiency virus infection were excluded (Figure 1).

### 2.2. Clinical Information

We retrieved the participants’ clinical information, such as age, gender, and details of the initial TB diagnosis from the respective hospital’s electronic records. Diagnosis date, anti-TB treatment, and date of diagnosis revision were reviewed in the hospital’s record which was the same as that of the Taiwan Centers for Disease Control because TB is a mandatory notification disease in Taiwan. The reasons for initial impression and diagnosis revision were recorded. Radiographic lesions of chest computed tomography (CT) scans were categorized as a nodule/mass, exudative lesion/consolidation, or pleural effusion (Figure 2). In addition, the presence of cavitation and the miliary radiographic pattern and location as well as the size of the main lesion were interpreted by pulmonologists in a default report form. The new diagnosis of lung cancer was defined by a pathology/cytology report and was responsible for the pulmonary lesion suspected to initially be TB. We also reviewed the methods to obtain evidence for cancer diagnosis, the stage of lung cancer, and the mortality.

### 2.3. Outcome and Statistical Analysis

We defined revising the diagnosis early (<1 month) from TB to lung cancer as the primary outcome and obtaining the pathology proof early (<1 month) for lung cancer as the secondary outcome. Inter-group differences were compared using the Mann–Whitney *U* test or one-way ANOVA for continuous variables, where appropriate, and the *Chi* square test for categorical variables. A Kaplan–Meier (KM) curve was used for plotting the time against event curve and was compared using the log rank test. We used Cox proportional hazard regression for analyzing time to diagnosis revision and time to pathology proof obtainment. The clinical, radiological, and microbiological relevant factors were included in the multivariable analysis and the final model was analyzed by the stepwise method. All analyses were performed in SPSS version 19.0 (SPSS Inc, Chicago, IL, USA). All Kaplan–Meier curves were plotted using the Prism software package (GraphPad Version 5.00, San Diego, CA, USA) and analyzed using the log rank test. A 2-tailed *p* value of <0.05 was considered significant. 

## 3. Results

### 3.1. Participant Demographics and Reasons for TB Suspicions

During the study period, a total of 6683 patients had been diagnosed in the study hospitals. Among them, the diagnoses of 978 (14.6%) and 45 (0.7%) had been revised to non-TB causes and lung cancer, respectively (Figure 1). The average age was 66 years and 71% were male (Table 1). Acid-fast stain (AFS) was positive for 20% and the lesion size was around 4.7 cm. Pleural effusion and radiological findings of cavitary and miliary patterns accounted 20%, 22%, and 11%, respectively. The reasons for tentatively diagnosed TB included suspicion of radiological findings (*n* = 23, 51%), positive AFS (*n* = 12, 27%), pathology suspicion for TB (*n* = 2, 4%), lymphocyte-predominant pleural effusion (*n* = 2, 4%), and other clinical suspicion (*n* = 8, 18%). Although all these patients were diagnosed with TB, anti-TB treatment was only applied only in 82%. The average length of time to obtain pathology and revise the diagnosis were 52.9 ± 74.4 days and 57.6 ± 44.7 days, respectively.

### 3.2. Final Diagnosis of Lung Cancer Status 

The pathology proof was obtained using CT-guidance in 14 patients (31%), bronchoscopy in 11 patients (24%), echo-guidance in 9 patients (20%), surgical procedure in 9 patients (20%), and pleural/pericardial effusion in 2 patients (4%). In regard to the cancer stage, there were 26 patients (58%) with metastases (stage 4) in the final diagnosis of lung cancer. In addition, 8 patients (18%) were stage 1, 3 patients (7%) were stage 2, and 8 patients (18%) were stage 3. Regarding the cancer type, 59% of patients had adenocarcinoma, 12% had squamous cell carcinoma, 9% had small cell carcinoma, 15% had poorly-differentiated carcinoma, 1 had pleural cancer, and 1 had sarcomatoid carcinoma. Nineteen patients (42%) died within 1 year follow-up.

### 3.3. The Factors Favored Early Revising Diagnosis

For patients with early diagnosis revision (*n* = 17, 32%) (Table 1), their age and demographics were similar with those with late revision except they had less cavitation pattern (6% vs. 32%, *p* = 0.040) and anti-TB treatment (65% vs 93%, *p* = 0.017) as well as lower time to obtain pathology (46.9 days vs. 56.5 days, *p* = 0.009). In multivariable Cox proportion analysis, early obtaining pathology was an independent factor associated with early diagnosis revision (Hazard ratio (HR): 2.079 (1.041–4.154), *p* = 0.038). Anti-TB treatment (HR: 0.363 (0.120–1.097), *p* = 0.073) and positive AFS (HR: 0.409 (0.146–1.148), 0.089) were borderline associated with diagnosis revision negatively (Table 2). The Kaplan-Meier (KM) curve also showed that early obtaining pathology could be significantly correlated with diagnosis revision earlier (Figure 3, *p* = 0.015). 

### 3.4. The Factors Associated with Early Obtained Pathology

We aimed to study the factors correlated with obtaining pathology proof early (*n* = 26, 58%). The patients with early obtained pathology had similar age, sex, positive proportion of sputum AFS, radiological pattern, and diagnosis evidence in comparison to those with late obtained pathology (Appendix A in Appendix A). In contrast, larger mass size (5.8 vs. 2.7 cm, *p* < 0.001), more miliary pattern (19% vs. 0%, 0.043), less anti-TB treatment (69% vs. 100%, *p* = 0.008), and shorter time to diagnosis revision (42.2 vs 78.6 days, *p* = 0.006) were found in the early pathology group than in late group.

We validated associated factors for early pathology by multivariable Cox proportional regression analysis (Table 3). The analysis showed that lesion size (HR: 4.258 (1.659–10.924) per 1 cm increment, *p* = 0.003), miliary radiographic pattern (HR:14.739 (4.096–53.038), *p* = 0.001), empirical anti-TB treatment (HR: 0.305 (0.094–0.994), *p* = 0.049), and sputum culture positive for nontuberculous mycobacteria (NTM) (HR: 0.310 (0.114–0.841), *p* = 0.021) were independent factors. The four independent factors were shown to be statistically significant by KM curves (Figure 4). 

## 4. Discussion

In the present study, the incidence of patients with misdiagnosis of TB was around 14.6% with some revised early to obtain a diagnosis of lung cancer (0.7%). We conducted this study to review the patients with initial notification of TB and then diagnosis revision to lung cancer in a TB-prevalent area. This study found that obtaining pathology early could help diagnoses be revised earlier. In particular, patients with a lesion size >3 cm, a radiologic miliary pattern, no anti-TB treatment, and cultures negative for NTM could alert clinicians to arrange a pathology exam. In contrast, those with a lesion size ≤3 cm, no radiologic miliary pattern, anti-TB treatment, and cultures positive for NTM are prone to have a delayed diagnosis and our conclusion emphasizes that lung cancer should be kept in mind in this subgroup and earlier pathology should be obtained if feasible. 

The clinical and radiological manifestations of lung cancer and tuberculosis are similar in some aspects [14,15]. For example, they are chronic and could have mass lesions, cavitation, and multiple pulmonary lesions [16,17]. Many lung abnormalities, such as nodules or masses discovered by chest radiography or CT scan, are considered suspicious enough to prompt immediate biopsy. In certain clinical situations, it may be desirable to further characterize a pulmonary nodule by imaging. Some investigations found that the CT characteristics of lung nodules or masses helped in the differentiation of benign and malignant pulmonary lesions [19]. However, some patients are at high risk of complications during invasive procedures, such as biopsy and bronchoscopy, so clinicians might try anti-TB treatment and follow the response to judge the diagnosis [7]. Therefore, TB could be misdiagnosed in a patient actually having lung cancer. However, this could lead to a delayed lung cancer diagnosis and late cancer treatment, which could lead to poorer outcomes [20,21,22]. In addition, anti-TB treatment would be unnecessary and its adverse effect and cost could be harmful for patients [7,23]. In such a misdiagnosis, how to revise a diagnosis early is very important, especially in a TB-prevalent area [24].

In the present study, early obtained tissue proof is a direct factor associated with shorter time to diagnosis revision to lung cancer; it is easy to understand this causal relationship. Similar findings have been found in our previous observation [18] but no study repeated the findings until the present investigation. Although the misdiagnosis rate decreased, the harm is still present and needs to be improved because it matters in delaying cancer diagnosis and influences prognosis. For patients with positive sputum AFS, the diagnosis revision might be postponed. Because concomitant cancer and infection is not uncommon [25,26], we would not totally exclude mycobacterial infection before we yielded negative cultures for *Mycobacterium tuberculosis,* which is time consuming [11]. In regard to patients with anti-TB treatment, the diagnosis revision is also late. This might be explained by the fact that the overall status is highly suspected to TB, so the diagnosis revision would not be made until the patient did not respond to the course of treatment. However, the treatment response for TB is usually not fast [27,28].

Therefore, to obtaining pathology early would be a key step to decrease the time to revise the diagnosis to lung cancer. There were four factors that significantly predicted early obtained pathology. First, the size of the radiological lesion is important because large lesion always alert clinician more and the treatment response would be easily to follow. Once it is clear the patient is unresponsive to treatment and the lesion is huge, i.e., >3 cm, the clinician will favor further tissue proof if no contraindications. Second, the miliary pattern can exist in both TB and lung cancer [29,30]. However, the miliary pattern in lung cancer usually shows bigger, multiple nodules and indicates terminal cancer with metastasis, which hints to physicians to think of cancer instead of TB. Third, no anti-TB regimen and cultures negative for NTM indicate a low chance of mycobacterial infection, so the tie to obtain pathology might be early in clinical practice. In fact, even if sputum culture is positive for NTM, the clinical colonization of NTM is not low [31], so the diagnosis needs to be re-considered seriously. In contrary, those with late obtained pathology require us to pay attention; they usually have small lesions, receive anti-TB treatment, have cultures positive for NTM, and no miliary pattern. However, the validation for the four-factor model is needed in the future.

Several limitations existed in this study. First, this is a retrospective study and different groups are not identical in demographics. In addition, no standard protocol was implemented for the procedure to obtain pathology. Among the technical part, radiology follow-up, CT acquisition protocol, and post-processing are missing. Slice acquisition, respiratory artifacts, and dose-length product (DLP) radiation dose all could have affected the results. All these elements would be required to determine the potential clinical impact of this study. Third, the case number is small, although we did review the cases from four medical referral centers. If the parent population focused on TB notification cases, it would underestimate the percentage of lung cancer mimicking TB.

In conclusion, we found there is small proportion (0.7%), but not a rare amount, of cases of lung cancer initially misdiagnosed as TB. Under such misdiagnosis, obtaining pathology early is a solution to revising the diagnosis early. For those with small lesions, radiological findings other than miliary pattern, receiving empirical anti-TB treatment, and sputum cultures positive for NTM, the time to obtain pathology is longer and we should carefully monitor the treatment response and dynamic lesion change in these patients. Further tissue proof is suggested, especially in patients without obvious treatment response or definite TB evidence. 

Ethics approval and consent to participate: The Research Ethics Committee of National Taiwan University Hospital approved this study (IRB No.: 201811047RINA).

## Figures and Tables

**Figure 1 jcm-08-00700-f001:**
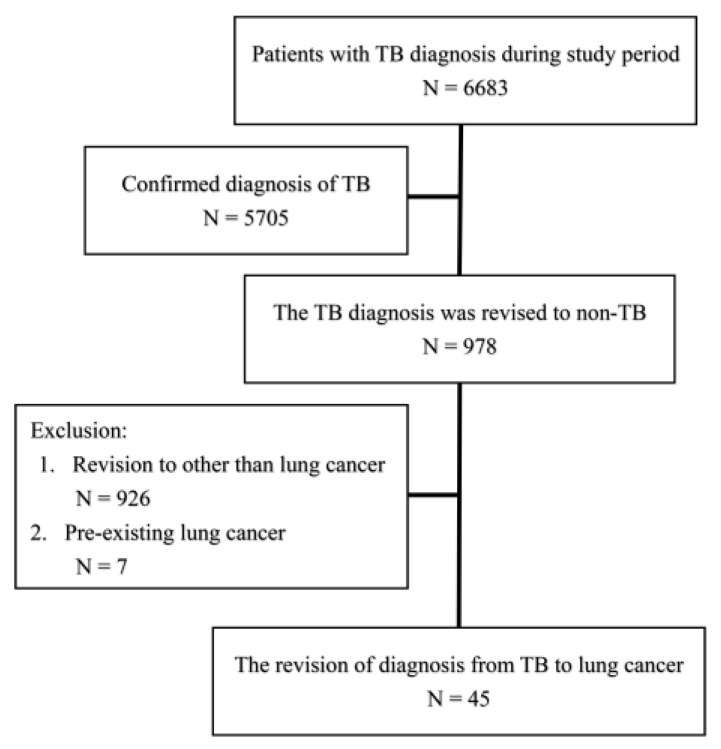
Flow chart of patient recruitment. TB, tuberculosis

**Figure 2 jcm-08-00700-f002:**
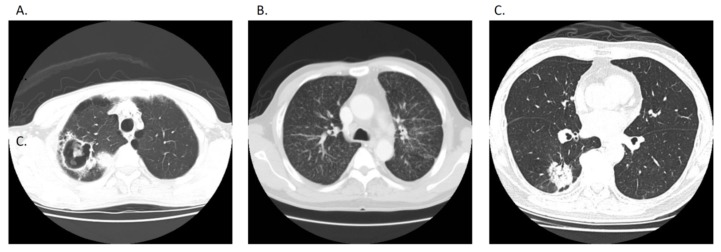
(**A**) Cavitary lung lesion, (**B**) bilateral multiple tiny lung nodules, and (**C**) consolidative lung patch, which all mimicked pulmonary tuberculosis infection [18].

**Figure 3 jcm-08-00700-f003:**
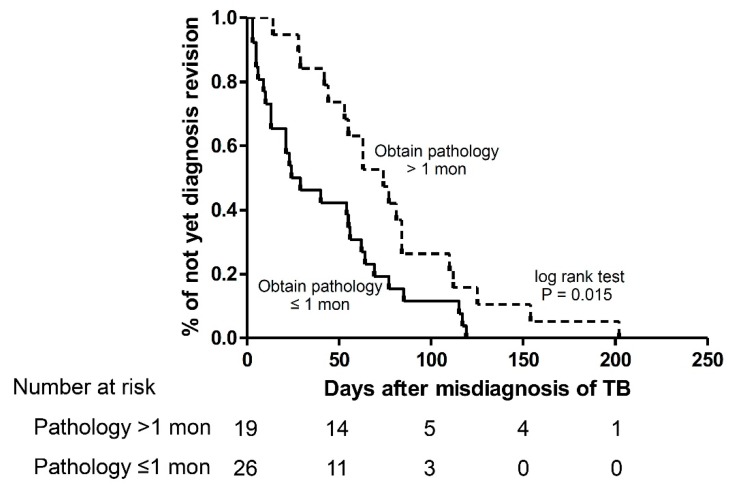
Kaplan Meier curve for time to revise diagnosis to lung cancer according to time to obtain pathology proof.

**Figure 4 jcm-08-00700-f004:**
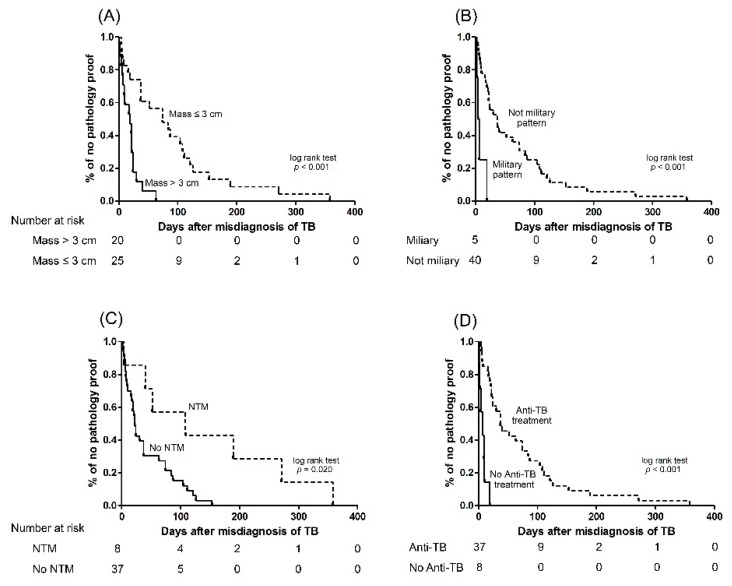
Kaplan–Meier curves for time to obtain pathology proof of lung cancer according to (**A**) lesion size, (**B**) radiographic miliary pattern, (**C**) mycobacterial culture, and (**D**) anti-tuberculosis treatment. NTM, nontuberculous mycobacteria; TB, tuberculosis.

**Table 1 jcm-08-00700-t001:** The demographics of patients according to the time to revise the diagnosis.

	All (*n* = 45)	Early Revision (*n* = 17)	Late Revision (*n* = 28)	*p* Value
Age (years)	66.1 ± 13.3	69.3 ± 11.8	64.1 ± 13.9	0.242
Male sex	32 (71%)	11 (75%)	21 (75%)	0.460
Microbiology				
AFS *				0.368
Positive	9 (20%)	4 (24%)	5 (18%)	
Negative	35 (78%)	12 (71%)	23 (82%)	
Culture *				0.147
NTM	8 (18%)	4 (24%)	4 (14%)	
Negative	16 (36%)	3 (18%)	13 (46%)	
Chest CT radiographic pattern				0.529
Nodular	28 (62%)	11 (65%)	17 (61%)	
Consolidation	15 (33%)	6 (35%)	9 (32%)	
Pleural effusion	2 (4%)	0	2 (7%)	
Lesion size (cm)	4.7 ± 2.8	5.2 ± 3.0	4.4 ± 2.7	0.235
Bilateral	6 (13%)	3 (18%)	3 (11%)	0.710
Cavitation	10 (22%)	1 (6%)	9 (32%)	0.040
Miliary pattern	5 (11%)	3 (18%)	2 (7%)	0.277
Pleural effusion amount				0.388
<1/3 hemithorax	3 (7%)	2 (12%)	1 (4%)	
1/3–2/3 hemithorax	5 (11%)	2 (12%)	3 (11%)	
>2/3 hemithorax	1 (2%)	1 (6%)	0	
The cause for TB suspicion				0.732
AFS (+)	12 (27%)	5 (29%)	7 (25%)	
Pathology suspicion	2 (4%)	0	2 (7%)	
Radiological suspicion	23 (51%)	9 (53%)	14 (50%)	
Clinical suspicion	8 (18%)	3 (18%)	5 (18%)	
Anti-TB treatment	37 (82%)	11 (65%)	26 (93%)	0.017
Days to obtain pathology	52.9 ± 74.4	46.9 ± 104.0	56.5 ± 50.7	0.009
Days to diagnosis revision	57.6 ± 44.7	15.1 ± 9.4	83.4 ± 37.0	<0.001

Abbreviations: AFS, acid-fast smear; CT, Computed Tomography; TB, tuberculosis; NTM, nontuberculous mycobacteria. * One case and 21 cases were not tested for AFS and mycobacteria culture, respectively.

**Table 2 jcm-08-00700-t002:** Multi-variable analysis for early diagnosis revision.

Characteristics	Multivariate	
	HR (95% C.I.)	*p* Value
Early obtaining pathology vs late group	2.079 (1.041–4.154)	0.038
Empirical anti-TB treatment vs none	0.363 (0.120–1.097)	0.073
AFS positive vs negative/not done	0.409 (0.146–1.148)	0.089

Abbreviation: AFS, acid-fast smear; TB, tuberculosis. All clinical, radiological, and microbiological factors were used for multivariable Cox proportional regression using stepwise methods.

**Table 3 jcm-08-00700-t003:** Multi-variable analysis for early obtained pathology.

Characteristics	Multivariate	
	HR (95% C.I.)	*p* Value
Miliary radiographic pattern vs. others	14.739 (4.096–53.038)	<0.001
Lesion size per 1 cm increment	4.258 (1.659–10.924)	0.003
Empirical anti-TB treatment vs. no treatment	0.305 (0.094–0.994)	0.049
Culture (+) for NTM vs. negative and not done	0.310 (0.114–0.841)	0.021

Abbreviations: TB, tuberculosis; NTM, nontuberculous mycobacteria. All clinical, radiological, and microbiological factors were used for multivariable Cox proportional regression using stepwise methods.

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
