# Peer review of "Factors for the Early Revision of Misdiagnosed Tuberculosis to Lung Cancer: A Multicenter Study in A Tuberculosis-Prevalent Area"

_jcm, 2019, doi:10.3390/jcm8050700_

Reviewer 1 Report

This article dials with an interesting topic, such as reviewing the patients with initial notification of TB and then diagnosis revisio to lung cáncer; and analyzing the factors associated with early revisión that could improve patient care.

- The case patients was small. They found that, during the study period, 45 (0,7%) of 6683 patients were finally diagnosed as lung cáncer.

- The authors conclude: In patients for suspicion for TB but no mycobacterial evidence lung cáncer should be kept in mind, and pathology needs to be obtained early.

- Although the topic is interesting, data are heterogeneous and, there was not a standard protocol for the procedure to obtain pathology. Also outlined by the authors in the study limitations.

Among the technical part, radiology follow up, CT acquisition protocol and post-processing are missing. Slice acquisition, respiratory artifacts, DLP radiation dose, all these could have affected the results. All these elements would be required to determine the potential clinical impact of this study.

- Retrospective research was approved by the Ethics Committee.

- English edition is needed.

- The main objective of the study needs more explanation.

- Table 1 is self explanatory, but there is no CT evidence. What does it mean radiographic pattern?

- Authors should clarify the imaging procedures for Clinical Information. They explain that radiographic lesions were categorized as nodule/…, etc, but they do not explain the imaging method: radiographs, CT... It should be explained in detail, and images should be included in the article. They only mention CT for the pathology proof (CT-guidance). For example, the figure 1 Flow chart can be acompanied by images with examples.

- The authors should indicate whether they are aiming to determine the potential for CT to be used as a monitor of response in clinical practice in these cases. In this context, it would have been useful for the paper to discuss whether CT could provide an indication of malignancy (early revisión to final diagnosis)

- Other comments:

Title: This should indicate better and better writen the main objective of the paper.

Discussion: What differentiates this paper from others? Why should it be expected to be an improvement over what is currently known?

Author Response

To the Reviewer 1

Comments and Suggestions for Authors

This article dials with an interesting topic, such as reviewing the patients with initial notification of TB and then diagnosis revisio to lung cáncer; and analyzing the factors associated with early revisión that could improve patient care.

- The case patients was small. They found that, during the study period, 45 (0,7%) of 6683 patients were finally diagnosed as lung cancer.

- The authors conclude: In patients for suspicion for TB but no mycobacterial evidence lung cancer should be kept in mind, and pathology needs to be obtained early.

- Although the topic is interesting, data are heterogeneous and, there was not a standard protocol for the procedure to obtain pathology. Also outlined by the authors in the study limitations.

Response: Thank you for your comments. By current education and practice under TB guideline, the rate of lung cancer being misdiagnosed as TB is small. In addition, a retrospective study design lacked a standard protocol. We are total agree with you that the data are heterogeneous due to above-mentioned factors. However, this study is truly important to emphasize that lung cancer should be kept in mind in the population being treat as TB without culture evidence. If feasible, early obtaining pathology could be helpful to revise the diagnosis earlier. We have added the description in the limitations of the discussion section. Thank you.

-Among the technical part, radiology follow up, CT acquisition protocol and post-processing are missing. Slice acquisition, respiratory artifacts, DLP radiation dose, all these could have affected the results. All these elements would be required to determine the potential clinical impact of this study.

Response: Thank you for your comments and suggestions. Although the details of the chest image have not been standardized and provided, we reviewed and analyzed the most important and objective radiographic features such as lesion size, and radiographic pattern, which showed good predicting value in the present study. We appreciate your suggestions and have added this point in the limitation in Discussion section (Paragraph 5, Line 219 – 222).

1.      Retrospective research was approved by the Ethics Committee.

Response: This study was approved by the Research Ethics Committee of National Taiwan University Hospital (IRB No.: 201811047RINA).

2.      English edition is needed.

Response: We will prepare the revised manuscript to an MDPI journal for English Editing.

3.      The main objective of the study needs more explanation.

Response: Thank you for your suggestions. Taiwan has an intermediate prevalence of TB. Differentiation of pulmonary TB from lung cancer can pose a great challenge to clinicians. However, there have been few studies on the timing of invasive diagnostic procedures for diagnosis of pulmonary TB [18], especially in TB-endemic areas. (INTRODUCTION, paragraph 2, Line 53 – 56).

4.      Table 1 is self-explanatory, but there is no CT evidence. What does it mean radiographic pattern?

Response: The radiographic pattern means Chest CT findings. We corrected it to Chest CT radiographic pattern in Table 1.

5.      Authors should clarify the imaging procedures for Clinical Information. They explain that radiographic lesions were categorized as nodule/…, etc, but they do not explain the imaging method: radiographs, CT... It should be explained in detail, and images should be included in the article. They only mention CT for the pathology proof (CT-guidance). For example, the figure 1 Flow chart can be acompanied by images with examples.

Response: Thank you for your comments and suggestions. We explained that radiographic lesions of chest computed tomography (CT) scan were categorized as nodule/mass, exudative lesion/consolidation, pleural effusion (Figure 2). (METHODS, Clinical Information, Line 82 – 84 and Figure 2).

6.      The authors should indicate whether they are aiming to determine the potential for CT to be used as a monitor of response in clinical practice in these cases. In this context, it would have been useful for the paper to discuss whether CT could provide an indication of malignancy (early revisión to final diagnosis).

Response: Thank you for your comments and suggestions.

   Many lung abnormalities as nodules or masses discovered by CXR or CT are considered suspicious enough to prompt immediate biopsy. However, in certain clinical situations, it may be desirable to further characterize a pulmonary nodule by imaging. Some investigations found that the CT characteristics of lung nodules or masses helped in the differentiation of benign and malignant pulmonary lesion [19]. (DISCUSSION, Paragraph 2, Line 179- 183).

(Ref 19. Lobrano MB. Partnerships in oncology and radiology: the role of radiology in the detection, staging, and follow-up of lung cancer. Oncologist 2006;11(7):774-9).

7.      Other comments:

Title: This should indicate better and better written the main objective of the paper.

Discussion: What differentiates this paper from others? Why should it be expected to be an improvement over what is currently known?

Response: Thank you for your comments and suggestions.

   We revised the title as “Factors for Early Revising Misdiagnosis of Tuberculosis to Lung Cancer: A Multicenter Study in A Tuberculosis Prevalent Area.” to fit the objective of earlier diagnosis revision in lung cancer being misdiagnosed as TB.

In addition, we conducted this study to review the patients with initial notification of TB and then diagnosis revision to lung cancer in a TB prevalent area. This study finally found that early obtaining pathology could revise the diagnosis earlier. In particular, patients with lesion size >3 cm, radiologic military pattern, no anti-TB treatment and culture negative for NTM could alert clinician to arrange pathology exam. By contrast, those with size ≤3 cm, no radiologic military pattern, anti-TB treatment and culture positive for NTM are prone to be delay diagnosis and our conclusion emphasize that lung cancer should be kept in mind in this subgroup and suggest earlier pathology obtaining if feasible. We have added the description and revised the discussion section accordingly. (DISCUSSION, Paragraph 1, Line 169-176).

Reviewer 2 Report

I congrats authors for conducting this study. The study presented by the authors is clinically relevant and I agree that lung cancer should be kept in mind for the patients with suspicion of TB. I find the study design appropriate. However, I have concerns regarding the KM curve in Fig.2 and Fig. 3B. Ideally, the KM curves should have a common starting point (100 or 1), in Fig. 2 and Fig. 3B that doesn’t seems to be the case. Therefore, following can be revised:

1) KM curves in Fig. 2 and Fig. 3B - explanation of why they don’t have a single starring point? 

2) Please provide p values for the log rank test. 

3) Please provide the survival tables along with KM curves. 

Author Response

To the Reviewer 2

Comments and Suggestions for Authors

I congrats authors for conducting this study. The study presented by the authors is clinically relevant and I agree that lung cancer should be kept in mind for the patients with suspicion of TB. I find the study design appropriate. However, I have concerns regarding the KM curve in Fig.2 and Fig. 3B. Ideally, the KM curves should have a common starting point (100 or 1), in Fig. 2 and Fig. 3B that doesn’t seems to be the case. Therefore, following can be revised:

Response: Thank you for your comments and suggestions. We have re-plotted the KM curves (Figure 2 and 3) using GraphPad Prism (version 5.0) and corrected the problem of starting point. We revised the methods/results section and added the survival table along with KM curves. Thank you.

1) KM curves in Fig. 2 and Fig. 3B - explanation of why they don’t have a single starring point?

Response: We have re-plotted the KM curves (Figure 2 and 3) using GraphPad Prism (version 5.0) and corrected problem of starting point accordingly.

2) Please provide p values for the log rank test.

Response: We have provided p values for the log rank test in the results sections and showed the p values on the Figure 2 and 3.

3) Please provide the survival tables along with KM curves.

Response: We have provided survival tables along with KM curves on the Figure 2 and 3.

Round  2

Reviewer 1 Report

Minor revisión and text editing (english) is needed.